IFT-UAM/CSIC-24-161
MPP-2024-222
November 15[th], 2024

# A note on the Noether–Wald
# and generalized Komar charges

*Tomás Ortín[1,a] and Matteo Zatti[1,2,b]*

[1]*Instituto de Física Teórica UAM/CSIC*
*C/ Nicolás Cabrera, 13–15, C.U. Cantoblanco, E-28049 Madrid, Spain*

[2]*Max Planck Institut für Physik*
*Boltzmannstrasse 8, 85748 Garching, Germany*

## Abstract

We propose a simple algorithm to compute the generalized Komar charge of any exactly gauge- and diffeomorphism-invariant theory.

[a]Email: `tomas.ortin[at]csic.es`
[b]Email: `zatti[at]mpp.mpg.de`

Conserved charges can be used to characterize the states and the evolution of physical systems. In this short note we describe how to compute the generalized Komar charge of any exactly gauge- and diffeomorphism-invariant theory. Our goal is to provide a simple algorithm which can be easily applied to spacetime configurations with Killing isometries (cf. Eq. (18)). This is particularly useful to determine efficiently fundamental identities in black hole thermodynamics and to prove non-existence theorems for gravitational solitons and boson stars (see for instance Refs. [1–12]).

In $d$-dimensional General Relativity, for each of the Killing vectors $k$ of the metric of a given vacuum solution one can construct a $(d-2)$-form charge $\mathbf{K}[k]$, the so-called *Komar charge* [13], which is closed on-shell[1]

$$d\mathbf{K}[k] \doteq 0 \, . \tag{1}$$

In asymptotically-flat spacetimes, the integral of this charge at spatial infinity ($S_\infty^{d-2}$) gives, up to normalization, the value of the conserved charge of the spacetime corresponding to the Killing vector: total mass/energy if $k$ is a Killing vector that generates time translations, etc. In General Relativity, the Komar charge coincides with the Noether–Wald charge associated with the invariance under diffeomorphisms generated by vector fields $\xi$, $\mathbf{Q}[\xi]$, evaluated over the Killing vector $k$, $\mathbf{Q}[k]$. This fact suggests that, in more general theories (with matter or with terms of higher order in the curvature) the Komar charge may also be given by the Noether–Wald charge evaluated on $k$, $\mathbf{Q}[k]$, since $\mathbf{Q}[\xi]$ can be constructed in any theory invariant under diffeomorphisms. This naive expectation turns out to be false in general. In order to understand why and what has to be done to construct an on-shell-closed 2-form (a *generalized Komar charge*), it is convenient to review the algorithm that leads to $\mathbf{Q}[\xi]$.

Let us consider a theory of gravity, described by the Vierbein $e^a$, coupled to a number of matter fields denoted generically by $\varphi$, whose dynamics is dictated by the action $S[e, \varphi]$. Under a generic infinitesimal variation of the fields

$$\delta S[e, \varphi] = \int \left\{ \mathbf{E}_a \wedge \delta e^a + \mathbf{E}_\varphi \wedge \delta \varphi + d\mathbf{\Theta}(e, \varphi, \delta e, \delta \varphi) \right\} , \tag{2}$$

where, by definition, $\mathbf{E}_a$ are the Einstein equations, $\mathbf{E}_\varphi$ are the equations of motion of the matter fields and $\mathbf{\Theta}(e, \varphi, \delta e, \delta \varphi)$ is the *symplectic prepotential*. If the theory is exactly invariant under diffeomorphisms and any other kind of gauge transformations,[2]

$$\delta_\xi S[e, \varphi] = - \int d\iota_\xi \mathbf{L} \, , \tag{3}$$

where $\iota_\xi \mathbf{L}$ indicates the interior product of the vector field $\xi$ with the $d$-form $\mathbf{L}$.[3] On the other hand, if we use, instead, the general infinitesimal variation Eq. (2) particularized

---

[1]We indicate with $\doteq$ those identities which are only satisfied on-shell.

[2]We exclude Chern–Simons and any other terms which are invariant up to total derivatives we do not want to deal with. We must take into account that isometries induce gauge transformations, as explained in Refs. [14–16].

[3]We are using differential-form notation. the rest of our conventions and notation can be found in Refs. [14, 17].

for infinitesimal diffeomorphisms, we get

$$\delta_\xi S[e, \varphi] = \int \left\{ \mathbf{E}_a \wedge \delta_\xi e^a + \mathbf{E}_\varphi \wedge \delta_\xi \varphi + d\mathbf{\Theta}(e, \varphi, \delta_\xi e, \delta_\xi \varphi) \right\} . \tag{4}$$

Comparing these two expressions we conclude that

$$\mathbf{E}_a \wedge \delta_\xi e^a + \mathbf{E}_\varphi \wedge \delta_\xi \varphi = dN , \tag{5}$$

for some $(d-1)$-form $N$ that vanishes on-shell. This is the content of Noether's second theorem. Then,

$$\delta_\xi S[e, \varphi] = \int d\mathbf{\Theta}' , \tag{6a}$$

$$\mathbf{\Theta}' \equiv \mathbf{\Theta}(e, \varphi, \delta_\xi e, \delta_\xi \varphi) + N , \tag{6b}$$

and, comparing again this expression with Eq. (3) one concludes that the $(d-1)$-form

$$\mathbf{J}[\xi] \equiv \mathbf{\Theta}' + \imath_\xi \mathbf{L} , \tag{7}$$

is closed off-shell for any vector field $\xi$

$$d\mathbf{J}[\xi] = 0 . \tag{8}$$

This implies the local existence of Noether–Wald $(d-2)$-form $\mathbf{Q}[k]$ defined, up to total derivatives, by

$$d\mathbf{Q}[\xi] = \mathbf{J}[\xi] . \tag{9}$$

Let us examine the right-hand side of this equation using the definition of $\mathbf{J}[\xi]$, Eqs. (7), and that of $\mathbf{\Theta}'$, Eq. (6b)

$$d\mathbf{Q}[\xi] = \mathbf{\Theta}(e, \varphi, \delta_\xi e, \delta_\xi \varphi) + N + \imath_\xi \mathbf{L} . \tag{10}$$

The second term in the right-hand side vanishes on-shell while the first vanishes if we find *Killing* (or *reducibility* [18]) parameters[4] $k$ such that

$$\delta_k e^a = \delta_k \varphi = 0 , \tag{11}$$

because the symplectic prepotential is linear on $\delta_\xi e^a$ and $\delta_\xi \varphi$. The third term only vanishes in some simple theories like General Relativity with no matter or with free, massless scalars, and, in general, we have to deal with the equation

$$d\left( \mathcal{O}_s \mathbf{Q}[k] \right) \doteq \mathcal{O}_s \imath_k \mathbf{L} , \tag{12}$$

where we have introduced the on-shell-setting operator $\mathcal{O}_s$ that evaluates the expression on its right over the solution $s$. Smarr formulas can be computed integrating this

---

[4]In general, the transformations $\delta_\xi$ depend on the vector field $\xi$ and other gauge parameters.

identity over hypersurfaces with boundary on the horizon and spatial infinity [19–22]. In Refs. [1,2] it was argued that the right-hand side of Eq. (12) is always a total derivative

$$\mathcal{O}_s \iota_k \mathbf{L} \equiv d\omega_k \, , \tag{13}$$

and we can define the *generalized Komar* $(d-2)$-*form charge*

$$\mathbf{K}[k] \equiv -\mathcal{O}_s \mathbf{Q}[k] + \omega_k \, , \tag{14}$$

which is closed on-shell

$$d\mathbf{K}[k] \doteq 0 \, . \tag{15}$$

At first sight, this construction may give a trivial $\mathbf{K}[k]$, but the explicit calculations performed in Refs. [1–12] proof otherwise. However, a closer inspection of the way in which those explicit calculations have been performed shows that what was computed in those references is, actually, and more precisely,

$$\iota_k \mathcal{O}_s \mathbf{L} \equiv d\omega_k \, , \tag{16}$$

where $\mathcal{O}_s \mathbf{L}$ is typically obtained from the trace of the Einstein equations. The difference between this definition of $\omega_k$ and the former is a total derivative that vanished on-shell

$$\iota_k \mathcal{O}_s \mathbf{L} - \mathcal{O}_s \iota_k \mathbf{L} = d\left(\omega_k - \mathbf{Q}[k]\right) \doteq 0 \, , \tag{17}$$

and it is, precisely, the total derivative of the generalized Komar charge. Thus, we arrive at the following prescription:

$$d\mathbf{K}[k] = [\iota_k, \mathcal{O}_s] \mathbf{L} \, . \tag{18}$$

Let us see how the prescription works in the simple example of the Einstein–Maxwell theory in $d$ dimensions. Its action is

$$S[e^a, A] = \frac{(-1)^{d-1}}{16\pi G_N^{(d)}} \int \left[ \star(e^a \wedge e^b) \wedge R_{ab} - \tfrac{1}{2} F \wedge \star F \right] \equiv \int \mathbf{L} \, , \tag{19}$$

and its equations of motion and symplectic prepotential, defined by

$$\delta S = \int \left\{ \mathbf{E}_a \wedge \delta e^a + \mathbf{E} \wedge \delta A + d\mathbf{\Theta}(e, A, \delta e, \delta A) \right\} \, , \tag{20}$$

are given by

$$\mathbf{E}_a = \iota_a \star (e^c \wedge e^d) \wedge R_{cd} + \tfrac{1}{2} \left( \iota_a F \wedge \star F - F \wedge \iota_a \star F \right) \, , \tag{21a}$$

$$\mathbf{E} = -d \star F \, , \tag{21b}$$

$$\Theta(e, A, \delta e, \delta A) = - \star (e^a \wedge e^b) \wedge \delta \omega_{ab} + \star F \wedge \delta A \,, \tag{21c}$$

where $\iota_c$ stands for $\iota_{e_c}$, where $e_c = e_c{}^\mu \partial_\mu$ and where we are ignoring the factor $(16\pi G_N^{(d)})^{-1}$ for the moment in order to get simpler expressions. A straightforward calculation using the explicit expression of the Einstein equation Eq. (21a) gives

$$(-1)^{d-1} \iota_k \mathbf{L} = \iota_k \star (e^a \wedge e^b) \wedge R_{ab} + (-1)^d \star (e^a \wedge e^b) \wedge \iota_k R_{ab} - \tfrac{1}{2} \iota_k F \wedge \star F - \tfrac{1}{2} F \wedge \iota_k \star F$$

$$= k^a \mathbf{E}_a - \iota_k F \wedge \star F + (-1)^{d-1} \star (e^a \wedge e^b) \wedge \iota_k R_{ab} \,. \tag{22}$$

The assumption of invariance under the diffeomorphism generated by $k$ implies the existence of the Maxwell and Lorentz momentum maps $P_k$ and $P_k{}^{ab}$ satisfying the momentum map equations

$$\iota_k R^{ab} = -\mathcal{D} P_k{}^{ab} \,, \tag{23a}$$

$$\iota_k F = -d P_k \,. \tag{23b}$$

The first equation is always solved by the *Killing bivector* $\mathcal{D}^a k^b$. Using these equations and integrating by parts, we find

$$(-1)^{d-1} \iota_k \mathbf{L} = k^a \mathbf{E}_a + d P_k \wedge \star F + (-1)^{d-1} \star (e^a \wedge e^b) \wedge \mathcal{D} P_{k\,ab}$$

$$= d \left[ - \star (e^a \wedge e^b) P_{k\,ab} + P_k \star F \right] + P_k \mathbf{E} + k^a \mathbf{E}_a \,,$$

so that

$$\mathcal{O}_k \iota_k \mathbf{L} \doteq d(-1)^d \left[ \star (e^a \wedge e^b) P_{k\,ab} - P_k \star F \right] = (\mathcal{O}_k \mathbf{Q}[k]) \,. \tag{24}$$

On the other hand, taking the trace of the Einstein equation Eq. (21a)

$$e^a \wedge \mathbf{E}_a = (d-2) \star (e^c \wedge e^d) \wedge R_{cd} - \frac{(d-4)}{2} F \wedge \star F$$

$$= (d-2) \left[ (-1)^{d-1} \mathbf{L} + \tfrac{1}{2} F \wedge \star F \right] - \frac{(d-4)}{2} F \wedge \star F \,, \tag{25}$$

$$= (d-2)(-1)^{d-1} \mathbf{L} + F \wedge \star F \,,$$

so

$$\mathbf{L} = \frac{(-1)^d}{d-2} F \wedge \star F + \frac{(-1)^d}{d-2} e^a \wedge \mathbf{E} \,. \tag{26}$$

Then,

$$
\begin{aligned}
\imath_k \mathcal{O}_s \mathbf{L} &= \imath_k \left[ \frac{(-1)^d}{d-2} F \wedge \star F \right] \\
&= \frac{(-1)^d}{d-2} \left[ \imath_k F \wedge \star F + F \wedge \imath_k \star F \right] .
\end{aligned}
\tag{27}
$$

We can use Eq. (23b) in the first term. In the second, profiting from the fact that we are working on-shell, we can use the dual Maxwell momentum-map equation[5]

$$
\imath_k \star F \doteq -d\tilde{P}_k \,,
\tag{28}
$$

and, integrating by parts and using the equations of motion and Bianchi identities, we have

$$
\begin{aligned}
\imath_k \mathcal{O}_s \mathbf{L} &= \frac{(-1)^{d-1}}{d-2} \left[ dP_k \wedge \star F + F \wedge d\tilde{P}_k \right] \\
&= d \left\{ \frac{(-1)^{d-1}}{d-2} \left[ P_k \star F + \tilde{P}_k F \right] \right\} .
\end{aligned}
\tag{29}
$$

Now,

$$
[\imath_k, \mathcal{O}_s] \mathbf{L} = d(-1)^{d-1} \left\{ \star(e^a \wedge e^b) P_{k\,ab} - \frac{(d-3)}{d-2} P_k \star F + \frac{1}{d-2} \tilde{P}_k F \right\} \,,
\tag{30}
$$

which coincides with the result found in Ref. [4].

# Acknowledgments

The work of TO and MZ has been supported in part by the MCI, AEI, FEDER (UE) grants PID2021-125700NB-C21 ("Gravity, Supergravity and Superstrings" (GRASS)) and IFT Centro de Excelencia Severo Ochoa CEX2020-001007-S. The work of MZ has been supported by the fellowship LCF/BQ/DI20/11780035 from "La Caixa" Foundation (ID 100010434). TO wishes to thank M.M. Fernández for her permanent support.

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
