# Peer review of "A note on the Noether-Wald and generalized Komar charges"

_SciPost Physics Core_

## Round 1 · Referee Report · Anonymous (Referee 1) · 2025-9-12

Report
The results in this short paper are probably correct. The derivation is unfortunately not entirely. There is no way that equation (5), which is correct, follows from (2) (3) and (4). (5) is an independent computation that involves integration by parts on the derivatives of the vector field $\xi$ and the Noether identities, yielding an explicit expression for the on-shell vanishing $N$. Subtracting (3) and (2) says that an expression for the canonical Noether charge is $i_\xi L+\Theta$. Subtracting this says that $d(N+\Theta+i_\xi L)=0$ which is (8).
Before considering the rest of the paper, I believe that a correct explanation of these first steps is important.
Before considering the rest of the paper, I believe that a correct explanation of these first steps is important.
Recommendation
Ask for minor revision

Author: Matteo Zatti on 2025-09-26 [id 5865]
(in reply to Report 1 on 2025-09-12)Dear referee,
Please find attached a detailed reply to the report, addressing the point you raised.
Kind regards,
Matteo Zatti
Attachment:
answertoreferee20250922.pdf

---

## Editorial Decision

resubmitted